# Profiling system-wide variations and similarities between Rheumatic Heart Disease and Acute Rheumatic Fever–A pilot analysis

Ranjitha Guttapadu[1], Nandini Prakash[2], Alka M[2], Ritika Chatterjee[3], Mahantesh S[4], Jayranganath M[5], Usha MK Sastry[5], Jayshree Rudrapatna Subramanyam[6], Dipshikha Chakravortty[3,7], Kalpana S. R[2]*, Nagasuma Chandra[1,8,9]*

**1** IISc Mathematics Initiative, Indian Institute of Science, Bengaluru, Karnataka, India, **2** Department of Pathology, Sri Jayadeva Institute of Cardiovascular Sciences and Research, Bengaluru, Karnataka, India, **3** Department of Microbiology and Cell Biology, Indian Institute of Science, Bengaluru, Karnataka, India, **4** Department of Microbiology, Indira Gandhi Institute of child health, Bengaluru, Karnataka, India, **5** Department of Paediatric Cardiology, Sri Jayadeva Institute of Cardiovascular Sciences and Research, Bengaluru, Karnataka, India, **6** Department of Microbiology, Kidwai Memorial Institute of Oncology, Bengaluru, Karnataka, India, **7** Adjunct Faculty, Indian Institute of Science Research and Education, Thiruvananthapuram, Kerala, India, **8** Department of Biochemistry, Indian Institute of Science, Bengaluru, Karnataka, India, **9** Centre for Biosystems Science and Engineering, Indian Institute of Science, Bengaluru, Karnataka, India

* kalpanasr@gmail.com (KSR); nchandra@iisc.ac.in (NC)

**Data Availability Statement:** The data discussed in this publication has been deposited in NCBI's Gene Expression Omnibus (GEO) and is accessible through the GEO Series accession number

## Abstract

Rheumatic heart disease (RHD) continues to affect developing countries with low income due to the lack of resources and effective diagnostic techniques. Understanding the genetic basis common to both the diseases and that of progression from its prequel disease state, Acute Rheumatic Fever (ARF), would aid in developing predictive biomarkers and improving patient care. To gain system-wide molecular insights into possible causes for progression, in this pilot study, we collected blood transcriptomes from ARF (5) and RHD (5) patients. Using an integrated transcriptome and network analysis approach, we identified a subnetwork comprising the most significantly differentially expressed genes and most perturbed pathways in RHD compared to ARF. For example, the chemokine signaling pathway was seen to be upregulated, while tryptophan metabolism was found to be downregulated in RHD. The subnetworks of variation between the two conditions provide unbiased molecular-level insights into the host processes that may be linked with the progression of ARF to RHD, which has the potential to inform future diagnostics and therapeutic strategies. We also found a significantly raised neutrophil/lymphocyte ratio in both ARF and RHD cohorts. Activated neutrophils and inhibited Natural Killer cell gene signatures reflected the drivers of the inflammatory process typical to both disease conditions.

## Author summary

Rheumatic Heart Disease (RHD), a neglected disease in developing countries that lack access to advanced health care, continues to plague many children and adolescents. While

GSE209591 (https://www.ncbi.nlm.nih.gov/geo/query/acc.cgi?acc=GSE209591).

**Funding:** This work was supported in part by funds from Rajiv Gandhi University of Health Sciences, Bangalore, Karnataka, India, to NC and SRK (vide: RGU/Adv,Res/CR/02/2017-18 dt 02/05/2017Project code 17C005B) and from the Department of Biotechnology Bioinformatics Centre support to NC. RG was supported by the Prime Minister's Research Fellowship (PMRF) awarded by the Ministry of Education, Government of India. RC was supported by the CSIR research fellowship. The funders had no role in study design, data collection and analysis, decision to publish, or preparation of the manuscript.

**Competing interests:** I have read the journal's policy and the authors of this manuscript have the following competing interests: NC is a co-founder of HealthSeq Precision Medicine, IISc campus and qBiome Research Private Limited, IITM which have no role in this manuscript. All other authors also declare that no competing interests exist.

it is known that Acute Rheumatic Fever (ARF) plays a causal role in RHD, only a subset of ARF patients progress to RHD. Understanding the transcriptome level differences between these two disease states would aid in understanding the progression of the prequel state and stage an intervention at an appropriate time to avert fatal consequences. This pilot study aimed to understand the pathway-level perturbations caused in individuals with either disease condition and explore their similarities and differences. Using sensitive network mining approaches and transcriptomics, we identified differentially expressed genes (DEGs) in highly perturbed condition-specific protein-protein interactions networks unique to RHD compared to ARF by filtering out DEGs altered with respect to healthy and clinical controls. We also report differences in clinical parameters in the disease states. The study shows great potential in developing tools such as predictive biomarkers to study the progression of RHD while also unraveling novel pathway-level perturbations.

## Introduction

Rheumatic heart disease (RHD) is a grave complication of acute rheumatic fever (ARF), a common outcome of untreated pharyngitis by beta-hemolytic group A streptococcal (GAS) infection. Annually, ~700 million cases of pharyngitis are causally associated with GAS worldwide [1]. ARF and RHD are autoimmune and inflammatory disorders–a sequel to GAS infection with a prevalence of 33.4 million cases annually, resulting in 10.5 million Disability-Adjusted Life Years lost and 2,88,348 deaths [2,3]. Most of these fatalities/disabilities occur in non-industrialized countries [1]. The most common age group affected by ARF and RHD is between 5 and 15 years [4]. Primary and secondary prevention by way of prompt antibiotic treatment of GAS pharyngitis and preventing recurrences of ARF in RHD patients have steadily helped reduce the disease burden in the developed world [5]. However, it continues to haunt various developing countries, including Africa, the Western Pacific, and India, plagued by difficulties due to limited healthcare resources and restricted access to early diagnosis and treatment [6].

A recent population-level study based on echocardiography profiles reported the pooled prevalence of RHD to be 26.1 per 1000 using the World Health Organization (WHO) criteria, which appears to be inversely related to a country's income level. Further, it showed that while about 60.7% of the clinically diagnosed cases remain stable, about 7.5% of clinical and 11.3% of sub-clinical cases progressed to definite RHD. [7,8]

Due to the lack of established markers for identifying disease progression in RHD, no recommendations for intervention exist. Host susceptibility factors by way of genetic predisposition, antibodies, T cell-mediated molecular mimicry, and pro-inflammatory cytokine responses are suggested to play a significant role in disease progression [9]. Co-infections with viruses like *Coxsackie B* have also contributed to the pathogenesis [10–12]. Polymorphism in genes such as TGF-β1, TNF-α, TLR2, and IL-1 Ra is associated with increased susceptibility to RHD [13,14]. An increased expression of Th17 cell-associated cytokines [15], reduction in the number of circulating Treg cells [16], cross-reactivity between oligoclonally expanded T-cells and the GAS M-protein and cardiac proteins such as myosin and laminin24 are among the factors associated with the pathogenesis of RHD [17].

Further, cellular adhesion molecules such as ICAM, VCAM and chemokines such as CCL3/MIP1α, CCL1/I-309, and CXCL9/Mig are upregulated in the valves and recruit auto-reactive T cells [18]. In addition, epigenetic changes initiated in the endothelial cells of the cardiac

valves following GAS infection mediated by Transforming growth factor β1 (TGF-β1) signaling are postulated to play an essential role in the onset and maintenance of the fibroinflammatory process seen in chronic RHD [19]. While there are some reports of variations in host response in ARF/RHD individually with respect to healthy subjects, attention has not been paid to the differences between the two, especially at a genome-wide level. Given the crucial role of genetic underplay in the disease conditions, we believed a transcriptomic-level understanding of the systemic variations in the disease conditions would aid in identifying factors associated with disease progression. As a primary objective, we sought an unbiased view of the transcriptomic profiles and related biological pathways between RHD and ARF to understand the differences and similarities between the two clinical conditions. As secondary objectives, we also aimed to uncover transcriptome and pathway level differences of the conditions with healthy and clinical controls. Knowledge of these variations is expected to inform efforts to develop more precise diagnostics and therapeutic strategies in the long run. This pilot study highlights the importance and immense potential of omics-level approaches for a global understanding of ARF and RHD and for identifying biomarkers suggestive of disease progression.

## Materials and methods

### Ethics statement

Sri Jayadeva Institute of Cardiovascular Sciences and Research's (SJICR) Institutional Medical Ethics Committee (vide no. SJICR/EC/2018/011 dt.23/02/2018) approved the study. Written informed consent for using samples for research was obtained from all the patients/guardians before sampling.

### Sample collection

Whole blood from RHD patients, congenital heart disease (CHD) patients, ARF patients, and healthy controls (HC) were collected at Sri Jayadeva Institute of Cardiovascular Sciences and Research, Bengaluru, Karnataka, India (Table 1). Although ideally, healthy children would be the best control group, considering that it was ethically and morally inappropriate to sample healthy children, blood from age-matched children with congenital heart disease (CHD) formed the clinical control group after confirming that they had no clinical, laboratory and echocardiographic evidence of coexisting ARF or RHD. Additionally, blood draws from healthy individuals over 18 years–(apparently in good health on clinical and laboratory evaluation) served as HC. The patients with RHD were visiting the outpatient department for follow-up at the time of recruitment and were all on ARF prophylaxis. None of the patients recruited for the study had any recurrence in the past six months. We took particular care to exclude

**Table 1. Demographic profile of the study population\*.**

| Variable | CHD (n = 30) | RHD (n = 30) | ARF (n = 17) |
|---|---|---|---|
| Age | 11.83±4.16 | 16.10±5.06 | 12.35±3.55 |
| Gender (M/F) | 15/15 | 15/15 | 10/07 |
| Height (Cm) | 134.98±17.99 | 154.56±13.95 | 146.44±20.19 |
| Weight (Kg) | 28.25±10.81 | 38.30±13.62 | 34.29±12.81 |
| BMI | 14.89±2.37 | 16.57±3.65 | 16.00±3.84 |

\* Note: Inclusive of sequenced samples

RHD cases with acute episodes to avoid sampling patients with acute over chronic RHD. This was done with the sole purpose of generating clean data without any ambiguity. The HC group served as a baseline reference for gene expression values, and patients with CHD served as a clinical control group. Patients fulfilling the major and minor modified Jones criteria [20] with echocardiographic changes were labeled ARF. Those with clinical findings and established cardiac valve damage in echocardiography were diagnosed as RHD. ARF and RHD patients reactive to HIV, HBV, HCV, and VDRL were excluded from the study. An additional criterion for including RHD patients in the study was ASO negativity and normal upper respiratory tract flora as per throat swab cultures. CHD patients with Atrial Septal Defect and Ventricular Septal Defect with a left to right shunt who were ASO negative and also negative for GAS on throat swab culture were included in the study. Blood from the two healthy subjects served as controls (HC) for the comparison of RNA sequencing results.

## RNA sequencing and pre-processing

RNA was extracted from the whole blood of 17 samples randomly selected from the respective groups (5 patients with ARF, 5 with RHD, 5 with CHD, and 2 HC subjects) and sequenced using the Illumina HiSeq X Ten platform after ensuring the RNA Integrity Number (RIN) of each sample was > 7. RIN values > 7 are considered acceptable in general [21]. Shah et al. 2019, also showed that RIN value > = 6.9 correlates with a fair $A_{260:280}$ ratio [22]. Further, >80% mapping to the well-established human reference genome was obtained for RNA sequences with RIN > 7 and thus was chosen as the threshold for selecting RNA. A HC group was included as a baseline reference for gene expression values. In addition, patients with CHD served as a clinical control group. The quality of the raw data obtained from sequencing was checked using the FastQC software (http://www.bioinformatics.babraham.ac.uk/projects/fastqc/). The reads obtained were trimmed using TrimGalore (https://www.bioinformatics.babraham.ac.uk/projects/trim_galore/) to remove the adapters attached. A Phred score of 30 was considered the threshold for quality scores. Raw data can be accessed using the accession number GSE209591 from the NCBI's Gene Expression Omnibus database. [23]

## DEG analysis and functional enrichment

RNASeq data analysis was carried out using Salmon for aligning, indexing, and quantifying the expression of transcripts [24] against the human reference genome GRCh38, followed by DeSeq2 [25] for downstream analysis where the differentially expressed genes (DEGs; 2-fold change, p-value < 0.05) in the gene expression profile of RHD patients compared to those of ARF patients were identified. DEGs were also computed between ARF vs. healthy, RHD vs. healthy, and CHD vs. healthy groups. The differential expression profiles between the groups were visualized using the pheatmap software (pheatmap (RRID:SCR_016418)). The biological pathways to which the DEGs belonged were identified by carrying out functional enrichment using the EnrichR software with default parameters [26].

## Network analysis

The general transcriptomics framework of identifying perturbations in a condition being studied via DEGs alone is highly dependent on sample sizes that skew the statistical significance of the findings. Further, genes do not act independently and are highly regulated by other genes. Thus, a network analysis approach that can identify relevant perturbations at the level of gene interactions is more beneficial. An in-house network mining approach that provides a global systems perspective of alteration in a condition was hence used to identify the most perturbed gene interactions and pathways. This approach has also been bench-marked against other

existing network analysis approaches such as WGCNA, ARACNE, and jActiveModules and was found to perform better in identifying pathway-level perturbations [27]. Fold changes obtained from the DEG analysis were mapped onto a well-curated human protein-protein interaction network that culminated from curating multiple regulatory and metabolic pathways from databases such as Kyoto Encyclopedia of Genes and Genomes (KEGG) and Omni-Path. A general version of the code used in these analyses can be accessed from Sambarey et al., 2017 and Sambaturu et al., 2021 (GitHub deposition -https://github.com/NarmadaSambaturu/PathExt, https://github.com/NarmadaSambaturu/EpiTracer) [28,29]. Briefly, condition-specific networks were obtained by integrating transcriptome data with the human protein-protein interaction network, and the top perturbed (combined active and repressed subnetworks with node weights as fold changed calculated as condition/control and control/condition respectively; and edge weights as an inverse square root of the product of the node weights of the nodes forming the edge) subnetworks ("TopNet") were computed, which reflect top-ranked connected sets of variations between two conditions being considered. The top-ranked paths are identified based on an all vs. all shortest path analysis using the Dijkstra's algorithm followed by ranking based on their path costs, equal to the sum of weights of each edge along the trace of the nodes in the path being traversed, normalized over path length. The networks were visualized using Cytoscape 3.8.2. Functional enrichment of the subnetworks was carried out using the EnrichR software with default parameters.

## Statistical analysis

Wald test was used for hypothesis testing in DeSeq2 to compare the test and control groups. The model fit generated by DESeq estimates a single dispersion parameter for each gene, given the size factor and condition groups that represents how far the observed count will be from the mean value of the model for a sample. The final dispersion value incorporates the within-group variability across all groups. This accounts for multi-group comparisons when all the samples are run together, and the contrast() function is used to identify group-wise DEGs. This approach was also followed in the study, where samples from all the groups were run together, and the contrast() function was utilized to identify DEGs from each pairwise group comparison. Statistically significant genes with a 2-fold change compared to their reference groups with p-value < 0.05 were considered DEGs. Further from the functional enrichment obtained through EnrichR, only pathways with a p-value < 0.05 were considered significantly perturbed. EnrichR uses Fisher's exact test. Statistically signification DEGs with Bonferroni correction at a threshold of adjusted p-value < 0.05 were also identified.

## Results

An overview of the study samples and the analysis carried out is shown in Fig 1. Briefly, patients were identified per the inclusion and exclusion criteria, and their blood transcriptomes were obtained using RNASeq technology. We aimed to identify the transcriptomic and pathway level differences and similarities between RHD and ARF that could warrant a large-scale clinical omics analysis to identify molecular markers to predict the progression of ARF to RHD, a highly neglected disease in developing countries. The primary comparison group was thus RHD vs. ARF. DEGs were determined by comparing whole blood transcriptomes of RHD vs. ARF and further analyzed using a genome-wide network analysis pipeline [28] to identify subnetworks capturing connected sets of most perturbed genes. DEGs obtained in group X vs. group Y indicate genes with altered regulation in group X using group Y as a reference. Further, to identify DEGs characteristic to the RHD vs. ARF pairwise comparison, secondary analyses were performed to identify DEGs in these conditions compared to their

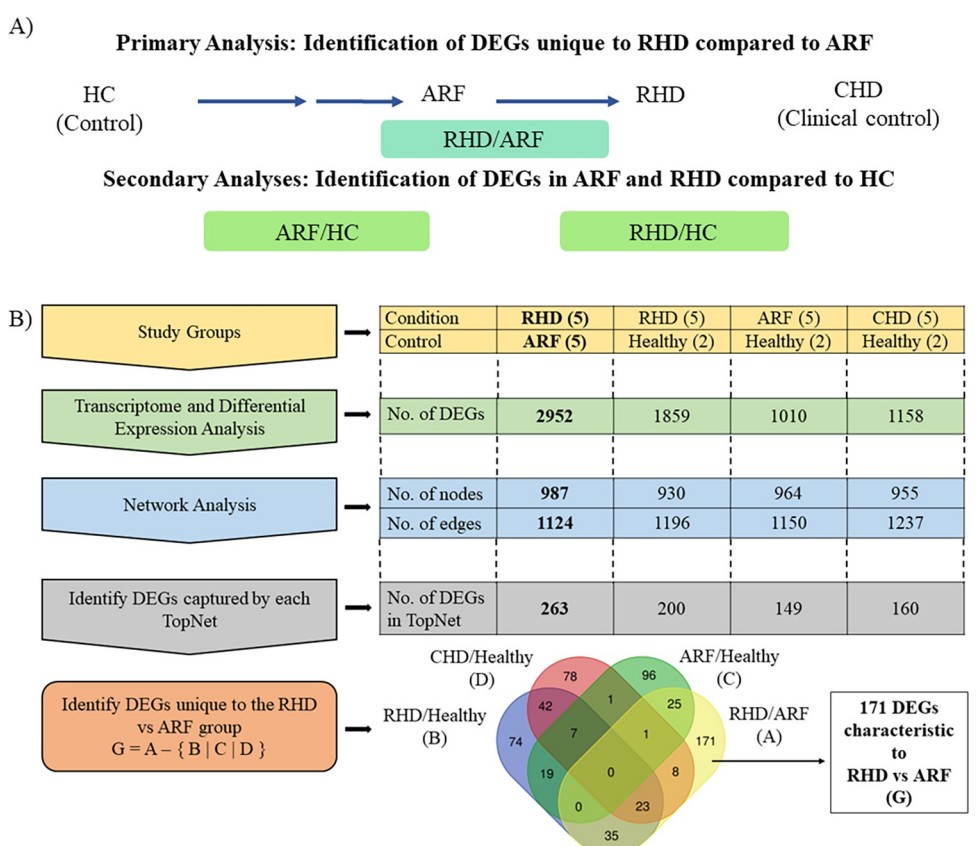

**Fig 1. Overview of the study.** (A) Relationship between the groups of analysis. (B) Overview of the workflow and results. Functionally significant DEGs characteristic to the RHD vs. ARF group obtained from a sensitive network mining approach were identified by excluding DEGs from sub-networks of RHD vs. Healthy, ARF vs. Healthy, and CHD vs. Healthy groups.

healthy controls to exclude them from the final candidates. Similarly, DEGs were also identified in the clinical control group (RHD). The DEGs subsequently captured in the respective subnetworks similarly produced were then excluded from those obtained in RHD vs. ARF subnetwork to filter out the DEGs characteristic to this pair finally.

## Peripheral blood profiles of patients

Various blood parameters observed in the patients are given in **Table 2**. Parameters from CHD cases were included as matched clinical controls. A significantly striking increase in neutrophils and thus the ratio of Neutrophils to Lymphocytes (NLR) was observed in patients with both ARF and RHD compared to CHD (P = 0.0012 and P = 0.0075 respectively, **Table 2**). Raised levels of ESR and CRP were also observed in ARF.

## Genome-wide perturbations between RHD versus ARF

2952 (646 upregulated and 2306 downregulated) DEGs were obtained between RHD vs. ARF (2-fold change, p-value <0.05; S1 File). Of these, 17 upregulated and 85 downregulated genes were also statistically significant with a Bonferroni correction with an adjusted p-value < 0.05 (S1 File). Functional enrichment analysis was performed on the DEGs, which identified pathways such as Chemokine signaling, Cadherin signaling, Wnt signaling, and TGF-β signaling to

**Table 2. Laboratory parameters of the study population\*.**

| Parameter | CHD (n = 30) | RHD (n = 30) | ARF (n = 17) |
|---|---|---|---|
| Neutrophils (%) | 48.48±8.73 | 58.75±12.22 | 59.68±12.28 |
| Lymphocytes (%) | 37.04±7.51 | 31.49±11.53 | 30.60±11.01 |
| Neutrophil-to-Lymphocyte ratio (NLR) | 1.39±0.48[a] | 2.46±2.06[b] | 2.44±1.55[c] |
| ESR (mm/hr) | 13.41±11.11 | 13.79±9.89 | 38.06±24.75 |
| CRP (mg/dl) | Not Done | 3.13±2.31 | 34.28±31.54 |
| ASO (IU/ml) | Negative | Negative | 536.87±199.28 |

\*Note: Inclusive of sequenced samples. a vs. b: P = 0.0075; a vs. c: P = 0.0012

be significantly perturbed. The top 10 DEGs in RHD (**Table 3**) were found to play an essential role in eicosanoid and leukotriene synthesis and arachidonic acid metabolism. These pathways have been proven critical in rheumatic and cardiovascular diseases [30].

Intriguing genes such as ALOX15 (Arachidonate 15-Lipoxygenase) and TIFAB (TRAF-interacting protein with FHA domain-containing protein B) were identified among the top-ranked DEGs. ALOX15 is a member of the lipoxygenase family that produces lipid mediators such as eicosanoids. Although ALOX15 has been implicated in cardiovascular disease genesis, it has not been previously associated with RHD [31]. Our analysis showed that ALOX15 was significantly upregulated in RHD compared to the ARF group. Similarly, TIFAB, a gene shown to be a susceptibility locus in patients with Kawasaki Disease for Coronary Artery Aneurysm [32], was another gene that was highly upregulated in RHD.

Next, to identify paths of perturbation that identify functionally significant DEGs, a network analysis was carried out as described in the methods to compute sub-networks of the highest perturbation between RHD and ARF. The RHD vs. ARF top perturbed network (Top-Net) consisted of 987 nodes and 1124 edges and was enriched with 263 DEGs with a p-value < 0.05 (out of which 13 also had an adjusted p-value < 0.05) (**Fig 2**). The active pathways enriched included well-established perturbations such as chemokine signaling [33], while the repressed pathways enriched included pathways such as tryptophan metabolism with a subset of tryptophan biosynthesis. Cardiovascular patients have been shown to have a low serum tryptophan concentration [34]. The top 10 most perturbed pathways based on p-value in the top active and top repressed networks are shown in **Table 4**.

## Genome-wide perturbations present in RHD and ARF compared to Healthy controls

A comparison of RHD and ARF, each with HC, was included to obtain perturbation profiles in each case. DEGs common to both were excluded in the pipeline to identify that

**Table 3. Top 10 significantly upregulated and downregulated genes in RHD compared to ARF (based on log₂FoldChange; p-value < 0.05).**

| Top 5 upregulated genes | | | Top 5 downregulated genes | | |
|---|---|---|---|---|---|
| Gene | log$_2$FoldChange | P-value | Gene | log$_2$FoldChange | P-value |
| MAGED4B | 7.2 | 0.00023 | ATP5MF-PTCD1 | -7.5 | 0.0052 |
| RHOXF1P1 | 4.3 | 0.00061 | USP17L15 | -7.1 | 0.0012 |
| ALOX15 | 4.2 | 2.28E-08 | CDR1 | -7.0 | 0.0032 |
| ARMC3 | 4.0 | 0.00094 | USP17L6P | -7.0 | 4.15E-05 |
| TIFAB | 3.9 | 1.05E-06 | USP17L22 | -6.9 | 0.014 |

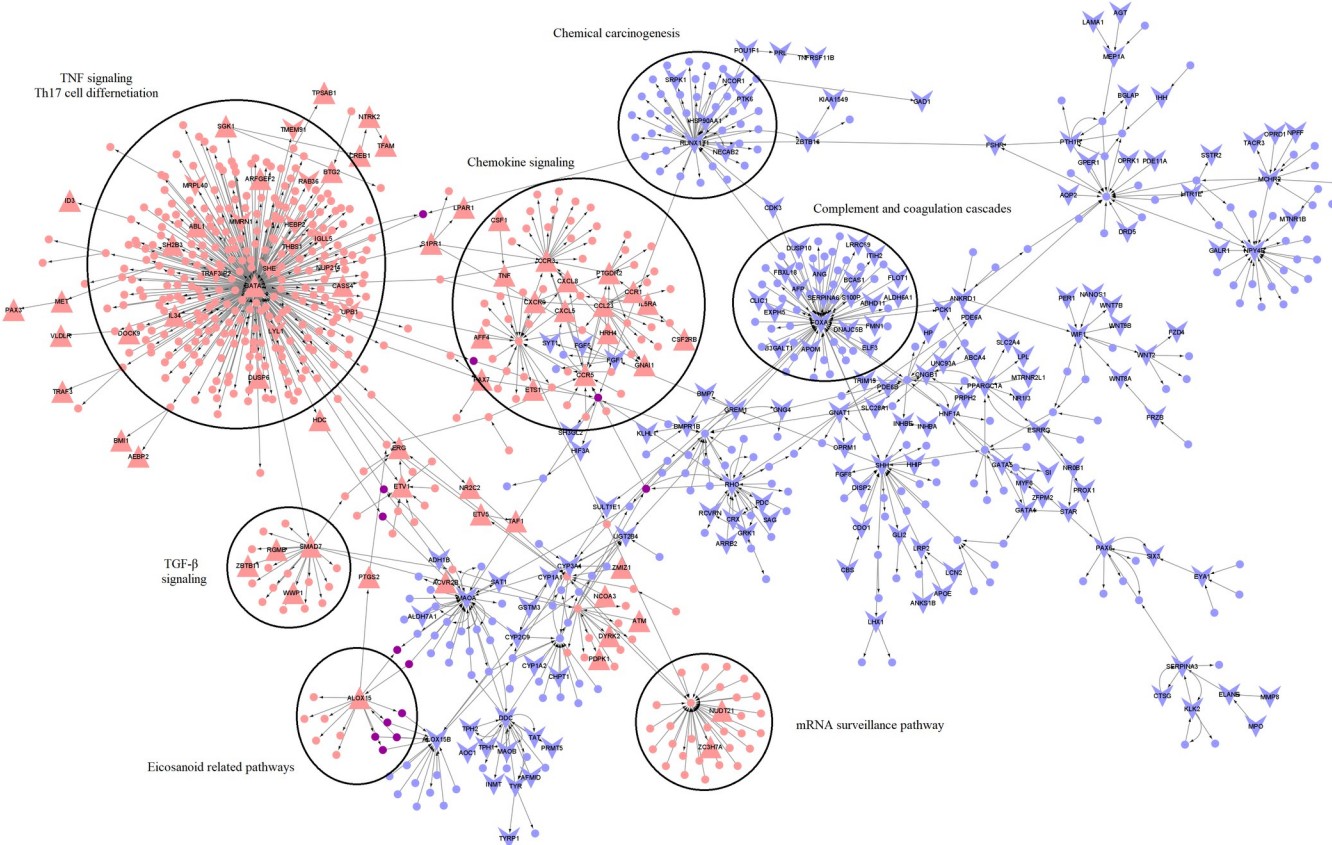

**Fig 2. TopNet for RHD vs ARF.** The red nodes represent genes in the top active network, the blue nodes represent nodes of the top repressed network, and the purple nodes indicate nodes common to both. Upper-facing triangles represent upregulated genes, and lower-facing triangles indicate nodes that are downregulated in RHD compared to ARF. Functional enrichment of some hub nodes such as GATA2, CCR3, SMAD7, and ALOX15 are also shown.

characteristic of RHD. DEGs common to RHD and ARF compared to HC are, however, helpful for understanding genome-wide perturbations that form the background on which both diseases arise. The data collected in this study allowed us to perform this comparison as a secondary analysis. DEGs identified in each condition against HC included: RHD vs. HC: 1859 DEGs (433 upregulated + 1426 downregulated with p-value < 0.05, out of which 1 upregulated and 3 downregulated genes also had an adjusted p-value < 0.05), ARF vs. HC: 1010 DEGs (795 upregulated + 215 downregulated with p-value < 0.05 out of which 11 upregulated and 2 downregulated genes also had an adjusted p-value < 0.05). In addition, transcriptomes were

**Table 4. Top 5 activated and top 5 repressed pathways based on p-value in RHD compared to ARF obtained using EnrichR.**

| Top 5 activated pathways | | Top 5 repressed pathways | |
|---|---|---|---|
| Term | P-value | Term | P-value |
| Chemokine signaling pathway | 1.57E-32 | Neuroactive ligand-receptor interaction | 1.88E-27 |
| Viral protein interaction with cytokine and cytokine receptor | 5.82E-30 | Tyrosine metabolism | 4.02E-20 |
| TNF signaling pathway | 8.36E-27 | Serotonergic synapse | 1.63E-19 |
| Cytokine-cytokine receptor interaction | 8.81E-27 | Tryptophan metabolism | 1.37E-18 |
| Pathways in cancer | 5.32E-25 | Chemical carcinogenesis | 9.45E-17 |

collected from the CHD group as clinical controls, and 1158 (164 upregulated + 994 downregulated with p-value < 0.05, of which 1 upregulated and 8 downregulated genes also had an adjusted p-value < 0.05) DEGs were identified. ARF and RHD conditions also showed several commonalities, with 254 DEGs in common (191 upregulated + 63 downregulated) compared to HC. These common DEGs were enriched for pathways such as carbon metabolism, PI3 Kinase pathway, Longevity regulating pathway, and Ras signaling.

Further network analysis was carried out to compute TopNets for ARF vs. HC (Fig 3), RHD vs. HC (Fig 4), and CHD vs. HC (Fig 5) groups. The RHD vs. HC TopNet consisted of 930 nodes and 1196 edges and captured 200 DEGs with a p-value < 0.05 (out of which 3 also

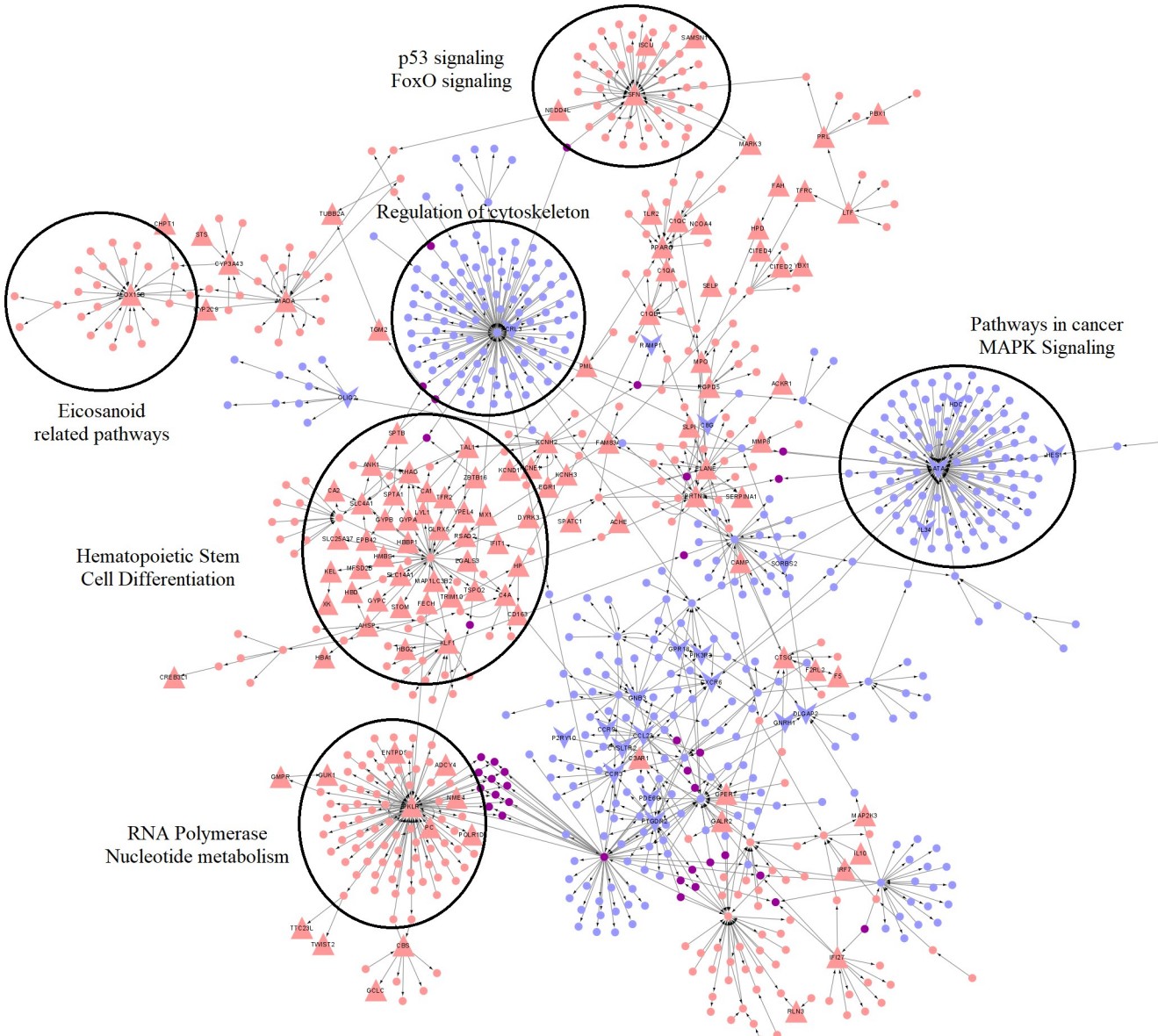

**Fig 3. TopNet for ARF vs. Healthy.** The red nodes represent genes in the top active network, the blue nodes represent nodes of the top repressed network, and the purple nodes indicate nodes common to both. The upper-facing triangles represent upregulated genes, and lower-facings ones indicate downregulated genes in each test condition compared to the reference groups. The DEGs are marked with a larger node size. Enrichment of some of the hub genes has also been shown.

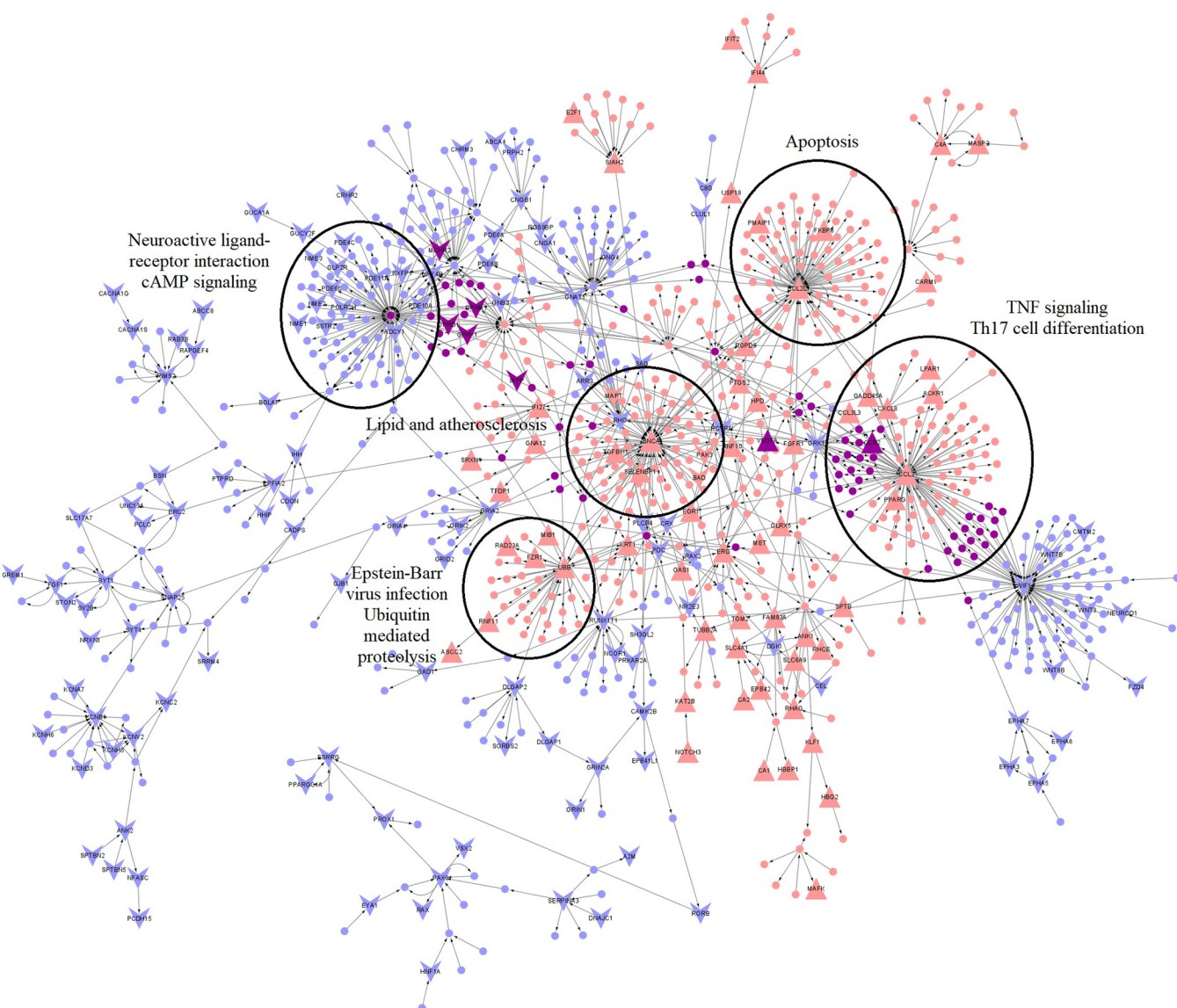

**Fig 4. TopNet for RHD vs Healthy.** The red nodes represent genes in the top active network, the blue nodes represent nodes of the top repressed network, and the purple nodes indicate nodes common to both. The DEGs are labeled and represented with larger node sizes. The upper-facing triangles represent genes that are upregulated, and lower-facing triangles indicate nodes that are downregulated in each of the test conditions compared to the reference groups. Enrichment of some of the hub genes has also been shown.

had an adjusted p-value < 0.05); on the other hand, the ARF vs. HC top perturbed network consisted of 964 nodes and 1150 edges and was able to capture 149 DEGs with p-value < 0.05 (out of which 2 also had an adjusted p-value < 0.05). The RHD TopNet nodes were enriched in pathways such as chemokine signaling and cAMP signaling pathway, while the ARF TopNet nodes indicated chemokine signaling and purine metabolism to be the most perturbed.

28 DEGs were common to the DEGs captured by the top perturbed networks of RHD and ARF compared to the healthy group. CHD vs. Healthy top perturbed network consisted of 955 nodes and 1237 edges and was able to capture 160 DEGs with p-value < 0.05 (out of which 5 also had an adjusted p-value < 0.05) and was also enriched in chemokine signaling.

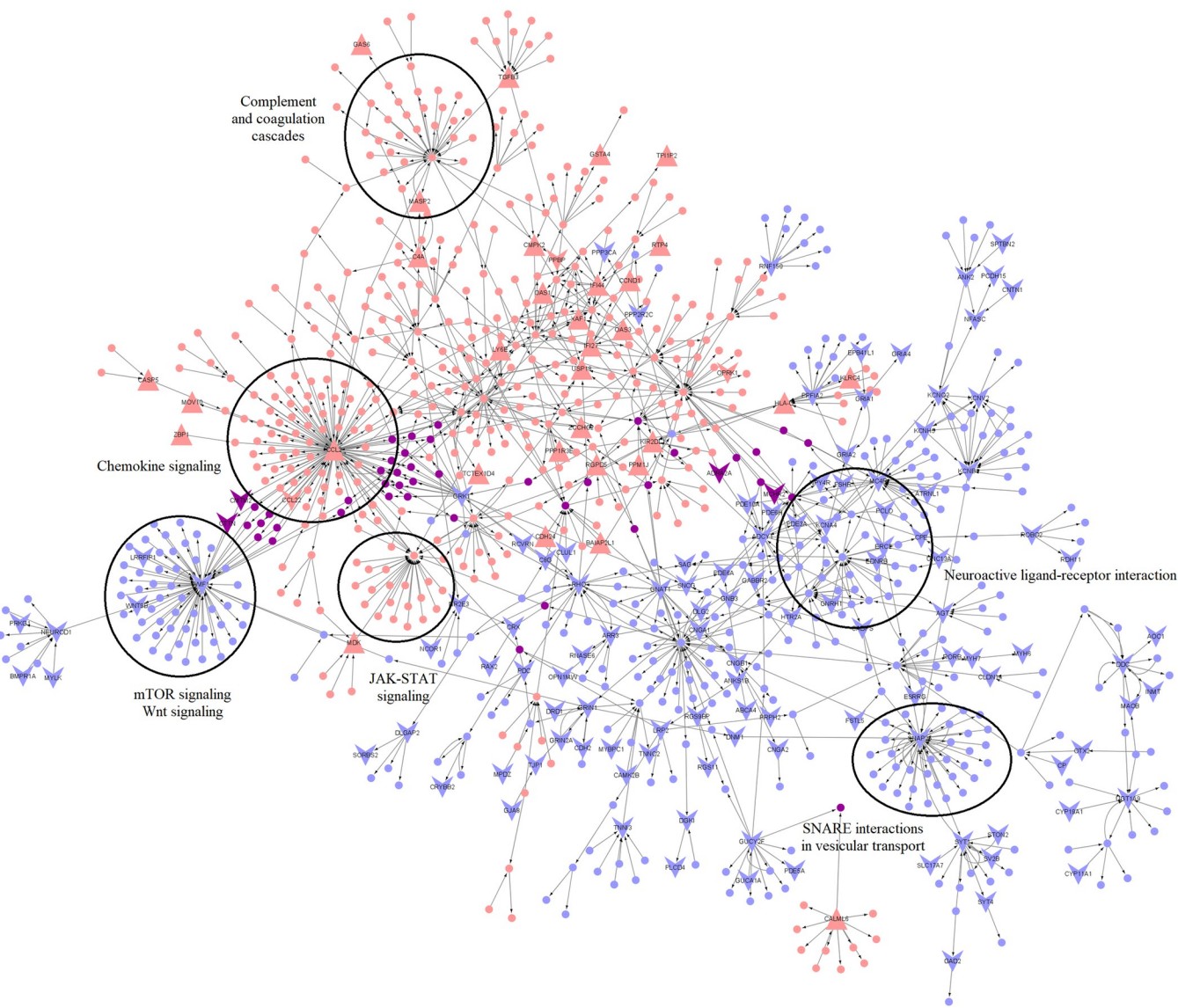

**Fig 5. TopNet for CHD vs Healthy.** The red nodes represent genes in the top active network, the blue nodes represent nodes of the top repressed network, and the purple nodes indicate nodes common to both. The DEGs are labeled and represented with larger node sizes. The upper-facing triangles represent genes that are upregulated, and lower-facing triangles indicate nodes that are downregulated in each of the test conditions compared to the reference groups. Enrichment of some of the hub genes has also been shown.

ARF and RHD are sequels to streptococcal infection and thus have an underlying inflammatory response that starts acutely in ARF and turns chronic in RHD. The inflammatory responses are regulated by chemokine, and cytokine signaling, such as increased production of IL6, which plays a role in CRP production, IL8, which is involved in chemotaxis, and TNFα which plays a role in increased production of pro-inflammatory cytokines, as well as chemotaxis in both the disease conditions. An increase in the production of these chemokines and cytokines eventually leads to valve damage [35]. This condition also reflects in CHD [36]. The current study's results also encapsulate these variations, indicating that the TopNets correctly capture the known immune responses in the disease conditions and their similarities.

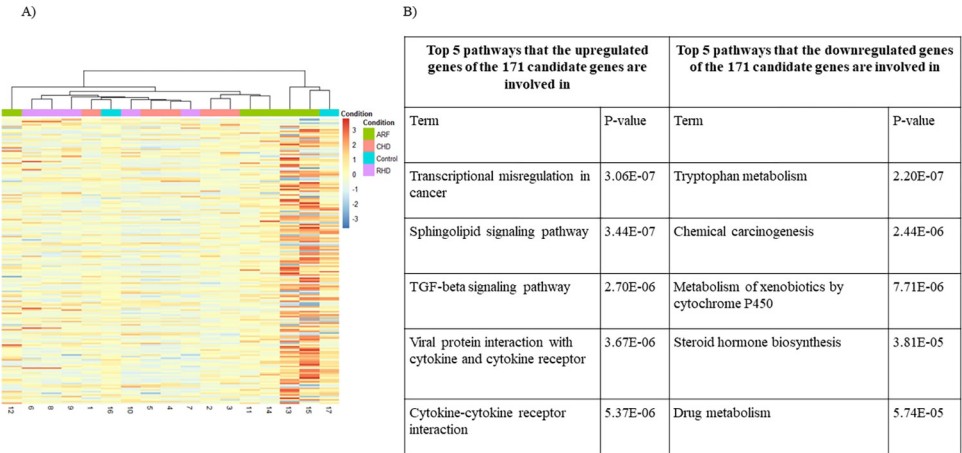

**Fig 6.** A) Heatmap of the 171 DEGs. Each row denotes a gene while each column represents a sample B) Top 5 pathways that the upregulated and downregulated genes of the 171 candidate genes unique to the RHD vs. ARF group are involved in (ranked based on p-value).

## Genome-wide perturbations specific to RHD vs. ARF (not present in RHD vs. HC, ARF vs. HC, or CHD vs. HC)

Identifying the overlap among all the DEGs captured by the TopNets yielded 171 DEGs (47 were upregulated and 124 were downregulated), unique to the RHD vs. ARF group (**Fig 1**). Some of these genes include chemokine and chemokine-related genes such as TNF, CCR1, CCR5, and genes involved in cytokine production such as SMAD7, indicating that our approach is capturing variations that are well established in the disease pathogenesis, including immunological perturbations. A heat map of normalized counts was then generated using the pheatmap software to hierarchically cluster the samples based on Z-scores scaled across the row (gene) for the 171 DEGs identified as characteristic of RHD. Hierarchical clustering groups samples together iteratively without prior knowledge of the sample group (unsupervised). Samples with similar expression profiles and, in turn, similar phenotypes cluster together and are represented using a dendrogram, where the branches represent the distance or closeness between the samples. In this set as well, when annotated, the ARF and RHD samples fairly clustered well. Although a larger cohort of samples would provide a better understanding, there seems to be a reasonable level of similarity among the RHD samples represented in purple (3 out of the 5 samples clustered together) and ARF samples represented in green (4 out of the 5 samples clustered together) indicating that the patients in these groups show a similar expression profile of this 171 gene set (**Fig 6A**). The complete list of 171 DEGs is given in **S1 File**.

Further, functional enrichment of the 171 DEGs was carried out to identify the perturbed biological pathways. This analysis indicated that the most significantly perturbed pathways are TGF-β signaling, tryptophan metabolism, cytokine-cytokine receptor interactions, and IL-17 signaling pathway (p-value <0.05) (**Fig 6B**). These pathways have been well studied in the context of cardiovascular diseases, thus theoretically validating the results obtained. The complete functional enrichment of nodes in each of the top perturbed networks and the 171 gene set is shown in **S2 File**. These 171 genes form a list of candidate genes to differentiate between RHD and ARF and could be studied in a larger cohort of patients whose disease status is tracked over time to narrow down further a gene signature that could provide insights into

characterizing features that lead to disease progression. Ultimately, such a gene panel would help predict the fate of the disease in patients and assist in choosing the right treatment strategy.

## Discussion

RHD continues to be prevalent in developing countries due to various factors, some of which are: socio-economic issues like inadequate housing leading to overcrowded dwellings, poor sanitation, poor nutrition, dearth of awareness about the disease, unhygienic living conditions, lack of access to primary health care, outdated cardiologic diagnostic tools in some remote areas and limited resources for early diagnosis and treatment [37]. As per the 2015 global disease burden estimates, 13.2 million people in India were afflicted with RHD, and ~1,20,000 succumbed to it. India has thus gained the ignominy or the dubious distinction of being labeled the RHD capital of the world [38]. School surveys over the past ten years indicate the mean prevalence of RHD to range from 0.5 to 1.47 per 1000 children aged 5–15 years. However, these figures may be underestimated, with the prevalence being much higher [39,40].

Recognizing the similarities and differences between the underlying genetic profiles of the two disease states would help understand the disease at a deeper level and provide pointers for identifying predictive biomarkers that indicate a probable progression. A raised neutrophil/lymphocyte ratio (NLR) indicates systemic inflammation and is directly proportional to rheumatic carditis severity. It has been proposed as a simple, helpful tool for assessing ARF patients on long-term follow-up [41,42]. Our results also conform to the high NLR observed in patients with ARF and RHD. While GAS is causally associated with ARF/RHD, a close collaboration with the host response is believed to be involved in valve damage.

Further, on those lines, activated neutrophils and macrophages are predicted to play a seminal role in the pathogenesis of this multi-component disease [43]. NLR and transcriptomic profile of the whole blood have been correlated in various disease conditions like cancer, infectious diseases, and autoimmunity [44–49]. A high NLR was also identified in our study's ARF and RHD samples. The transcriptomics and network analyses carried out in this study also reflected the implication of this clinical finding where multiple proinflammatory genes, such as CXCL8 and CXCR6, were upregulated in RHD and ARF. Further pathway enrichment of the TopNets also captured Neutrophil extracellular trap formation as a significantly perturbed pathway (S2 File). Based on the results of the present study, we propose that peripheral blood cell population subsets and transcriptomic signatures could also be used as biomarkers of disease severity in ARF/RHD. Various pro-inflammatory molecules and extremely toxic cationic substances released during NETosis have been predicted to destroy valves synergistically [43]. Contextually, it was intriguing to find increased expression of neutrophilic granule contents like ELANE, ARG1, DEFA, and MMP8 amongst the commonly upregulated genes in both ARF and RHD, which was perhaps indicative of the cells' activation status and may be envisaged to contribute to tissue injury [50]. This may be particularly so considering that neutrophils operating via their various granule contents (such as MMP8) have increasingly been recognized to play a prominent role in the etiopathogenesis of various other diseases, including cardiovascular diseases [51]. A corresponding downregulation of Natural Killer (NK) cell-activating receptors reflecting a low activation status was also observed in both the disease groups in the present study. The activation status of NK cells is the net result of the activating and inhibitory signals generated during cell-cell interaction through their surface receptors [52].

Further, an inverse correlation between NLR and NK cell activation has been reported where deficient NK cell activation indicates a high inflammatory status and vice versa [53,54]. Overall, our surmise is that activated neutrophils may form a common footing for tissue injury

in the etiopathogenesis of ARF and RHD. Thus, we consider whole blood transcriptomics more relevant and representative of systemic involvement vis-à-vis transcriptomic profiling of peripheral blood mononuclear cells in ARF and RHD.

Using high-throughput RNA Sequencing data and a sensitive network mining approach, we identified 171 genes as candidate genes that may help differentiate RHD and ARF. Some of these genes were involved in pathways that play an essential role in RHD pathogenesis, such as TGF-β signaling, tryptophan metabolism, cytokine interactions, and chemokine signaling. Based on the increased expression of the fibrogenic cytokine—TGF-β1 [55,56] and its down-stream molecules in the resected valves of RHD, Karthikeyan et al. have succinctly posited the fundamental role of TGF-β1 signaling in the initiation and sustenance of fibro-inflammatory process in the disease. Increased secretion of TGF-β1 along with pro-inflammatory cytokines and type I interferons in the local milieu of the valve following GAS infection is envisaged to activate the TGF-β- SMAD3/4 signaling cascade leading to valvular fibrosis [19]. Activation of this pathway has also been corroborated in a rat model of RHD [57]. 10 out of the 171 DEGs we found unique to RHD vs ARF are involved in TGF-β signaling, out of which ID3, RGMB, TNF, ACVR2B, SMAD7 were upregulated, and INHBA, BMPR1B, THBS1, BMP7, INHBE were downregulated. Although some of these are positive regulators, some negative regulators of TGF-β signaling seem to be upregulated as well, indicating some amount of downregulation of the pathway in RHD. This may indicate that the anti-inflammatory TGF-β signaling in the peripheral blood leukocytes that is generally upregulated in ARF when compared to healthy individuals may reduce upon progression to RHD. In essence, the reduced anti-inflammatory activity of the peripheral blood leukocytes observed in RHD in our study may contribute to the proinflammatory process in the damaged valves as evidenced by reduced TGF-β and increased TNF alpha signaling in this disease state. Increased expression of TGF-β and the downstream signaling pathway has been demonstrated in cardiac valves in RHD both in humans and experimental animals [55–57]. Thus, a crucial role of augmented secretion of TGF-β locally in the cardiac valves by the activated tissue macrophages steering upregulated TGF-β signaling in the valvular interstitial cells during ARF has been hypothesized to aid in the progression of the disease to RHD. Hence, in a nutshell, TGF-β signaling in the lesion is envisaged to initiate and sustain the fibroinflammatory process resulting in endothelial to mesenchymal transformation, thus promoting fibrosis and increased stiffness of the valves, and peripheral blood leukocytes may be aiding the proinflammatory process. Overall, our results point to a scenario of complex remodulation of the TGF-β signaling pathway occurring in the peripheral blood leukocytes, which may promote inflammation in the valves [19]. Analysis in larger sample size would aid in clearly profiling the trend of expression of the pathway and its effect on ARF progression to RHD.

Further, low plasma tryptophan has been reported in ARF patients with recurrent streptococcal infections [58]. Our analysis identified a subgroup of six DEGs out of 171 DEGs characteristic of the RHD vs. ARF group that are known to play a role in tryptophan metabolism (a pathway that houses tryptophan biosynthesis as a sub-pathway), including THP1, THP2, AFMID, CYP1A1, CYP1A2 and ALDH7A1 to be downregulated indicating that tryptophan metabolism may get further downregulated during the progression of ARF to RHD. Tryptophan metabolism genes were, however, not amongst the most commonly dysregulated genes in ARF and RHD when compared to healthy individuals in the present study. Simultaneous estimation of amino-acid plasma levels and a larger sample size would help clarify this aspect. We also observed increased expression of complement factor 4A (C4A) in the peripheral blood leukocytes in both the disease states. While the liver is the predominant source of component C4A of the complement, peripheral blood leucocytes are known to be extrahepatic sites of synthesis of C4A [59]. Previous proteomics data about plasma levels of this

complement component are divergent—one stating increased concentrations and another observing the reverse [60–62]. Although increased mRNA expression in the blood vs. increased plasma protein levels of C4a may not mean the same in its entirety–increased synthesis vs. increased release of C4a as a result of activation of complement pathway by the classical/lectin pathway; future large-scale studies with simultaneous estimation of plasma levels of various complement components would be required to resolve this issue. Also, while C4A is known to be an anaphylatoxin and a mediator of local inflammation, the relevance of the increased expression of C4A observed in our study is unknown.

Altered T cell function and cytokine production are also considered to play a vital role in RHD severity. Some key cytokines linked to RF and RHD pathogenesis and maintenance include IL-6, IFN-γ, TNF-α, IL10, IL12, and IL17 [63–65]. The 171 gene candidates found in our study remarkably capture these known variations. These gene candidates also included chemokines such as CXCL8, TNF, and CCR5 which have been reported to be high in RHD [33].

The promising findings from this study are preliminary and limited by the small sample size, as patient and disease level heterogeneity may skew the observed results. A larger cohort of around 20–30 samples per group would help provide more confidence in the effects observed. Furthermore, the findings can lead to the design of an expanded clinical study to get statistically significant molecular profiles that differ in the two conditions, which upon experimental validation, would aid in developing a clinically translatable gene signature to monitor the progression of ARF to RHD. Given the preliminary findings, the study brings to light the remarkable potential of omics approaches to uncover intricacies of disease progression and the ability to tease apart closely related disease classes.

Overall, this pilot study identifies similarities and differences between transcriptomic profiles of ARF and RHD patients and provides a glimpse of the significant perturbations in the disease conditions when compared to each other and healthy as well as clinical controls. It led to identifying 171 gene candidates that show potential to be predictive biomarkers of progression from ARF to RHD, highlighting the need for a large-scale omics study to acquire an in-depth understanding of the field. The study also hints at the critical role of raised NLR, neutrophil activation, and simultaneous NK cell suppression as a common ground for the evolution of both the disease states as well as genes and pathways perturbed in RHD compared to ARF that reflect changes due to disease progression, and paves the way for designing further experiments to gain mechanistic insights. We further highlight whole blood transcriptomics as more relevant and representative of systemic variations of peripheral blood cell subsets in ARF and RHD.

## Supporting information

**S1 File. DEG list.** List of DEGs across all the conditions and DEGs captured in each TopNet.
(XLSX)

**S2 File. Functional Enrichment.** Functional enrichment of genes of each TopNet and the 171 shortlisted gene candidates.
(XLSX)

## Acknowledgments

We would like to thank Dr. Sowmya and Dr. Bharath, Department of Paediatric Cardiology, Sri Jayadeva Institute of Cardiovascular Sciences and Research, Bengaluru, Karnataka, India, for their help in recruiting study cases. We truly acknowledge the DBT-IISc partnership umbrella program for advanced research in biological sciences and Bioengineering to DC and

infrastructure support from ICMR (Centre for Advanced Study in Molecular Medicine), DST (FIST), DST-TATA Innovation Fellowship, and UGC (special assistance).

## Author Contributions

**Conceptualization:** Jayshree Rudrapatna Subramanyam, Dipshikha Chakravortty, Kalpana S. R, Nagasuma Chandra.

**Data curation:** Ranjitha Guttapadu, Nandini Prakash, Jayshree Rudrapatna Subramanyam, Kalpana S. R, Nagasuma Chandra.

**Formal analysis:** Ranjitha Guttapadu, Nandini Prakash, Jayshree Rudrapatna Subramanyam, Kalpana S. R.

**Funding acquisition:** Dipshikha Chakravortty, Kalpana S. R, Nagasuma Chandra.

**Investigation:** Ranjitha Guttapadu, Nandini Prakash, Alka M, Ritika Chatterjee, Mahantesh S.

**Methodology:** Ranjitha Guttapadu, Nandini Prakash, Alka M, Jayshree Rudrapatna Subramanyam, Dipshikha Chakravortty, Kalpana S. R, Nagasuma Chandra.

**Project administration:** Kalpana S. R, Nagasuma Chandra.

**Resources:** Ranjitha Guttapadu, Nandini Prakash, Alka M, Mahantesh S, Jayranganath M, Usha MK Sastry, Jayshree Rudrapatna Subramanyam, Dipshikha Chakravortty, Kalpana S. R, Nagasuma Chandra.

**Software:** Ranjitha Guttapadu, Nagasuma Chandra.

**Supervision:** Kalpana S. R, Nagasuma Chandra.

**Validation:** Ranjitha Guttapadu, Ritika Chatterjee, Dipshikha Chakravortty, Nagasuma Chandra.

**Visualization:** Ranjitha Guttapadu, Jayshree Rudrapatna Subramanyam, Kalpana S. R, Nagasuma Chandra.

**Writing – original draft:** Ranjitha Guttapadu, Jayshree Rudrapatna Subramanyam, Kalpana S. R, Nagasuma Chandra.

**Writing – review & editing:** Ranjitha Guttapadu, Jayshree Rudrapatna Subramanyam, Kalpana S. R, Nagasuma Chandra.

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
