## [Decision Letter · Decision Letter 0]

12 Oct 2022

Dear Dr. Chandra,

Thank you very much for submitting your manuscript "Profiling system-wide variations and similarities between Rheumatic Heart Disease and Acute Rheumatic Fever" for consideration at PLOS Neglected Tropical Diseases. As with all papers reviewed by the journal, your manuscript was reviewed by members of the editorial board and by several independent reviewers. In light of the reviews (below this email), we would like to invite the resubmission of a significantly-revised version that takes into account the reviewers' comments. 

We cannot make any decision about publication until we have seen the revised manuscript and your response to the reviewers' comments. Your revised manuscript is also likely to be sent to reviewers for further evaluation.

Sincerely,

Liesl Joanna Zuhlke

Section Editor

Stuart Blacksell

Section Editor

Reviewer's Responses to Questions

**Key Review Criteria Required for Acceptance?**

**Methods**

-Are the objectives of the study clearly articulated with a clear testable hypothesis stated?

-Is the study design appropriate to address the stated objectives?

-Is the population clearly described and appropriate for the hypothesis being tested?

-Is the sample size sufficient to ensure adequate power to address the hypothesis being tested?

-Were correct statistical analysis used to support conclusions?

-Are there concerns about ethical or regulatory requirements being met?

Reviewer #1: Manuscript PNTD-22-01009

Profiling system-wide variations and similarities between Rheumatic Heart Disease and Acute Rheumatic Fever

The present article is well written and interesting. The authors describe the identification of a sub network comprising the most significantly and differentially expressed genes and most perturbed pathways in RHD compared to ARF.

They evaluated the transcriptoma of 5 patients presenting ARF and 5 presenting RHD and identified a subnetwork comprising 53 significantly differentially expressed genes and most perturbed pathways in RHD compared 54 to ARF. 

Figure 1, describe the overview of the article and is well presented.

Reviewer #2: The objectives are stated although only in non-specific terms. The sample size is very small rendering this a pilot analysis which needs to be clearly stated. While that in itself is not problematic there is very limited description of the clinical state of the patients and controls which will markedly influence the results (particularly given the small sample size). There is insufficient description of approaches to deal with multiple testing. Ethical review is described.

**Results**

-Does the analysis presented match the analysis plan?

-Are the results clearly and completely presented?

-Are the figures (Tables, Images) of sufficient quality for clarity?

Reviewer #1: Analysis, are correct, results are clearly and Figure is well presented

Reviewer #2: The results are moderately well presented but there is room for substantial improvement. The figures need to be provided in high resolution format as they are not currently readable. There needs to be much greater clarity of the relationships between the various tests performed i.e. RHD vs ARF, RHD vs HC etc. The Venn diagram provides some help with this but it would be useful to have a flow diagram with references back to this. It should be made much clearer which of the analyses was primary and which were secondary ideally with more stringent thresholds for secondary analyses. The content on the neutrophil-to-lymphocyte ratio seems out of keeping with the wider transcriptomics and further effort should be made to link these together to improve the flow of the manuscript as well as explain whether this work was part of the initial analysis plan.

**Conclusions**

-Are the conclusions supported by the data presented?

-Are the limitations of analysis clearly described?

-Do the authors discuss how these data can be helpful to advance our understanding of the topic under study?

-Is public health relevance addressed?

Reviewer #1: Yes for all questions

Reviewer #2: The manuscript would benefit from a clear summary that outlines the key findings in relation to the researchers' hypothesis. A substantially greater proportion of the discussion should be devoted to limitations, particularly in relationship to sample size and the clinical characteristics of the cohort and how this is might influence the results. At the current sample size the results are extremely preliminary and so the conclusions should be substantially more cautious and so relating the results to pathogenesis is less informative at this stage. A more useful message would be how given these primarily findings 'omics approaches might be useful to the field more broadly and what sample sizes would be necessary for definitive studies.

**Editorial and Data Presentation Modifications?**

Reviewer #1: None

Reviewer #2: In addition please address the following issues:

Introduction: 

 - Line 110 - correct prevalence in the Noubiap et al paper is 26.1 per 1000

Methods: 

- Line 145 - how were healthy controls selected and what was the justification for using these specific groups?

 - were either the ARF or RHD patients acutely unwell at the time recruitment? if so describe including timing from onset, if not describe time since last episode/recurrence

- Line 162 - define RIN abbreviations, if RNA integrity number justify why this threshold and not a higher value was used

- Line 177 - given the multi-way comparisons please explain what steps were taken to account for multiple testing

- Line 183 - for the in-house mining approach please justify use of this approach (as opposed to others) and describe availability of code (so that the work can be reproduced)

Results 

- Line 242 - Table 3 and Figure 2 - directionality needs to be made substantially clear in text, table and figures to indicate whether unregulated refers to up in RHD and down in ARF or vice-a-versa.

- Line 245 - remove "In addition to the well-known ones" and describe genes without assumption about which are or are not well-known by readers

- Line 264 - all network Figures must be provided in a high-resolution format that is easily readable 

- Line 303 - further justify statement about similarities between conditions - e.g. chemokine signalling is mentioned in relation to RHD, ARF, and CHD

- Line 321 - is the statement relating to the heatmap justified and are differences meaningful? although three RHD patients are neighbours the ARF patients appear scattered and the closest relationships appear to between different phenotypes. what parameters were used to draw this heatmap and how are they justified?

Discussion 

- Line 357 - explain relationship between neutrophil-lymphocyte ratio to the wider transcriptome findings -- are there prior studies transcriptomic studies of the neutrophil-lymphocyte ratio in other traits that inform or contradict findings?

- Line 366 - explain why up regulation of genes related to neutrophils is not a non-specific reflection of the acute state of patients enrolled in the study 

- Line 387 - relate text on TGF-beta to direction of up/down regulation in relation to RHD vs ARF - does this fit with the hypothesis ? 

- Line 396 - clarify reference 47 

- Line 407 - C4A is predominantly expressed in the liver -- please clarify the implications of this for the results

**Summary and General Comments**

Reviewer #1: The present article is well written and interesting. The authors describe the identification of a sub network comprising the most significantly and differentially expressed genes and most perturbed pathways in RHD compared to ARF.

They evaluated the transcriptoma of 5 patients presenting ARF and 5 presenting RHD and identified a subnetwork comprising 53 significantly differentially expressed genes and most perturbed pathways in RHD compared 54 to ARF. 

Figure 1, describe the overview of the article and is well presented.

Reviewer #2: This is a useful analysis in a neglected area but it should be presented as a pilot with very cautious results --- I would suggest the word "pilot" is added to the title. There needs to be much greater effort to identify key messages from the various comparisons made together are informative as well as indicate which of these findings are the most statistically robust given the multiple testing. It is vital that substantially greater clarity is added to the direction of effects and mandatory that the figures are presented in an updated high resolution readable format.

PLOS authors have the option to publish the peer review history of their article (what does this mean?). If published, this will include your full peer review and any attached files.

Reviewer #1: Yes: Luiza Guilherme

Reviewer #2: Yes: Tom Parks
---

## [Decision Letter · Decision Letter 1]

22 Mar 2023

Dear Dr. Chandra,

We are pleased to inform you that your manuscript 'Profiling system-wide variations and similarities between Rheumatic Heart Disease and Acute Rheumatic Fever – A pilot analysis' has been provisionally accepted for publication in PLOS Neglected Tropical Diseases.

Best regards,

Liesl Joanna Zuhlke

Section Editor

Stuart Blacksell

Section Editor

Thank you for extensive and comprehensive revisions. Please correct the remaining reference prior to acceptance.

Reviewer's Responses to Questions

**Key Review Criteria Required for Acceptance?**

**Methods**

-Are the objectives of the study clearly articulated with a clear testable hypothesis stated?

-Is the study design appropriate to address the stated objectives?

-Is the population clearly described and appropriate for the hypothesis being tested?

-Is the sample size sufficient to ensure adequate power to address the hypothesis being tested?

-Were correct statistical analysis used to support conclusions?

-Are there concerns about ethical or regulatory requirements being met?

Reviewer #1: The authors answered all the questions raised by some of the reviewers.

As mentioned before they describe the identification of a sub network comprising the most significantly and differentially expressed genes and most perturbed pathways in RHD compared to ARF.

Reviewer #2: The authors have done an excellent job of addressing my previous comments and including the issues raised above. The description of the study population is substantially improved. Thank you for including the terms pilot study as suggested. Details of ethical and consent approval provided.

**Results**

-Does the analysis presented match the analysis plan?

-Are the results clearly and completely presented?

-Are the figures (Tables, Images) of sufficient quality for clarity?

Reviewer #1: Yes.

Results are clear and well presented. Figures and Tables 1 to 5 presented good quality and clarity.

Reviewer #2: The description of the results including the figures is improved and satisfactory.

**Conclusions**

-Are the conclusions supported by the data presented?

-Are the limitations of analysis clearly described?

-Do the authors discuss how these data can be helpful to advance our understanding of the topic under study?

-Is public health relevance addressed?

Reviewer #1: Yes, as mention by the authors " the preliminary findings of the study brings to light the remarkable potential of omics approaches to uncover intricacies of disease progression and the ability to tease apart closely related disease classes.

Overall, this pilot study identifies similarities and differences between transcriptomic profiles of ARF and RHD patients and provides a glimpse of the significant perturbations in the disease conditions, when compared to each other and healthy as well as clinical controls ".

Reviewer #2: The discussion and conclusions are improved including the section on limitations.

**Editorial and Data Presentation Modifications?**

Reviewer #1: (No Response)

Reviewer #2: Reference 58 still needs correcting. I believe the citation you are after is: Kurup RK, Kurup PA. Endogenous hypodigoxinemia-related immune deficiency syndrome. Int J Neurosci. 2003 Sep;113(9):1287-303. doi: 10.1080/00207450390212294. PMID: 12959745. (https://pubmed.ncbi.nlm.nih.gov/12959745/)

**Summary and General Comments**

Reviewer #1: As mentioned above the article is interesting, well presented and considered all ethics aspects.

See also review comments in the attachment.

Reviewer #2: Overall I think this is now a useful contribution to the ARF / RHD literature and I am delighted that the authors have invested the time needed to improve it and I would recommend it is accepted. The only very minor issue that is outstanding from my previous comments (I am afraid) is the citation for reference 58.

PLOS authors have the option to publish the peer review history of their article (what does this mean?). If published, this will include your full peer review and any attached files.

Reviewer #1: **Yes: **Luiza Guilherme

Reviewer #2: **Yes: **Tom Parks

---

## [Editor Report · Acceptance letter]

31 Mar 2023

Dear Dr. Chandra,

We are delighted to inform you that your manuscript, "Profiling system-wide variations and similarities between Rheumatic Heart Disease and Acute Rheumatic Fever – A pilot analysis," has been formally accepted for publication in PLOS Neglected Tropical Diseases.

Best regards,

Shaden Kamhawi

co-Editor-in-Chief

Paul Brindley

co-Editor-in-Chief
